# Care for Women with Delusional Disorder: Towards a Specialized Approach

**Alexandre González-Rodríguez** [1,*], **Mary V. Seeman** [2], **Aida Álvarez** [3], **Armand Guàrdia** [3], **Nadia Sanz** [3], **Genís F. Fucho** [3], **Diego J. Palao** [1] and **Javier Labad** [4]

1   Department of Mental Health, Parc Tauli University Hospital, Autonomous University of Barcelona (UAB), I3PT, CIBERSAM, Sabadell, 08280 Barcelona, Spain; dpalao@tauli.cat
2   Department of Psychiatry, University of Toronto, 605 260 Heath Street West, Toronto, ON M5T 1R8, Canada; mary.seeman@utoronto.ca
3   Department of Mental Health, Parc Tauli University Hospital, Sabadell, 08280 Barcelona, Spain; aalvarezp@tauli.cat (A.Á.); aguardia@tauli.cat (A.G.); nsanzl@tauli.cat (N.S.); gffucho@tauli.cat (G.F.F.)
4   Department of Mental Health, Consorci Sanitari del Maresme, CIBERSAM, 08304 Mataró, Spain; jlabad@csdm.cat
*   Correspondence: agonzalezro@tauli.cat

**Abstract:** Delusional disorder is a difficult-to-treat clinical condition with health needs that are often undertreated. Although individuals with delusional disorder may be high functioning in daily life, they suffer from serious health complaints that may be sex-specific. The main aim of this narrative review is to address these sex-specific health needs and to find ways of integrating their management into service programs. Age is an important issue. Delusional disorder most often first occurs in middle to late adult life, a time that corresponds to menopause in women, and menopausal age correlates with increased development of both somatic and psychological health problems in women. It is associated with a rise in the prevalence of depression and a worsening of prior psychotic symptoms. Importantly, women with delusional disorder show low compliance rates with both psychiatric treatment and with medical/surgical referrals. Intervention at the patient, provider, and systems levels are needed to address these ongoing problems.

**Keywords:** psychosis; delusional disorder; women; health care; sex-specific treatment

## 1. Introduction

For several decades, epidemiological studies have consistently reported gender differences in the expression of mental disorders [1]. The prevalence of some common mental disorders such as mood disorders, anxiety and somatoform disorders are reported to be substantially higher in women than in men while substance use disorders are higher in men [1,2]. In general, male/female ratios in the frequency and severity of psychiatric problems vary with age [2]. For instance, during adolescence, girls are more likely than boys to show depressive symptoms and eating disorders, to present clinically with suicidal ideation, and to attempt suicide [3]. In adulthood, the prevalence of depressive and anxiety disorders remains higher in women than in men, while the number of men with substance use disorders substantially outstrips that of women. On the other hand, in schizophrenia and bipolar disorder, no consistent gender differences in prevalence have been noted [3,4]. While the occurrence rate of bipolar disorder is similar in the two sexes, women are more likely to seek help from mental health services than men are [5]. Ages of onset often vary. In schizophrenia, psychotic symptoms, on average, start a few years later in women than in men [6].

Prevalence and onset age aside, in the context of many psychotic illnesses, optimal care for women differs from that for men [7]. Some of the critical issues regarding sex-specific health needs involve

social status in one's society, ease of access to quality health care, exposure to reproductive casualty and the stresses of parenting and care taking. There are also inherent differences between males and females with respect to pharmacological response to treatment or to endocrine, metabolic, immune, and cardiovascular vulnerabilities [7,8].

Although comparably rich research has been focused on various aspects of health in women with other illnesses, little is known about the specific health needs of women with delusional disorder (DD). Because DD is a schizophrenia-related disorder, many gender-specific health needs may prove similar to those that apply to schizophrenia but as mentioned above, sex-specific needs change with age and the two disorders, DD and schizophrenia, characteristically affect women of different ages [3,8].

In this paper, we cover a gap in the clinical/academic literature by focusing on the specific health needs of women with DD. At the end, we will propose a program of service that focuses on the comprehensive care of these women.

The aim of this narrative, non-systematic review is to address some important questions concerning the management of women with DD that places special importance on sex-specific health needs. The main questions to be addressed are: (A) Do women with DD have particular health needs different from those of men? (B) Do women with DD need sex-specific monitoring of potential comorbidities? (C) What are the specific recommendations for clinicians caring for women with DD? (D) Is a gender-specific focus reasonable for women with DD?

## 2. Method

Electronic searches were conducted through the PubMed database for English, Spanish, German or French papers that referred in their titles or abstracts to gender and health needs in patients with DD. Papers were only included if they were focused on health outcomes (physical illness) or described the occurrence of physical comorbidities in DD populations.

The following keywords were used: women AND (health OR care) AND psychosis. Of papers that were retrieved, those published in the last decade were first included; classic papers frequently cited were subsequently also added. Google Scholar was also searched to add relevant papers in the field not found in the PubMed database. The screening and selection processes were undertaken by AGR and AA. Several hundred titles and abstracts were scanned; most were excluded as they did not address health needs in patients with DD. Figure 1 shows the methodological procedure and results from the selection process. In some cases, full-text documents of selected abstracts were not available. After screening all accessible full-text papers, a total of 65 records were identified as relevant to our questions (Figure 1).

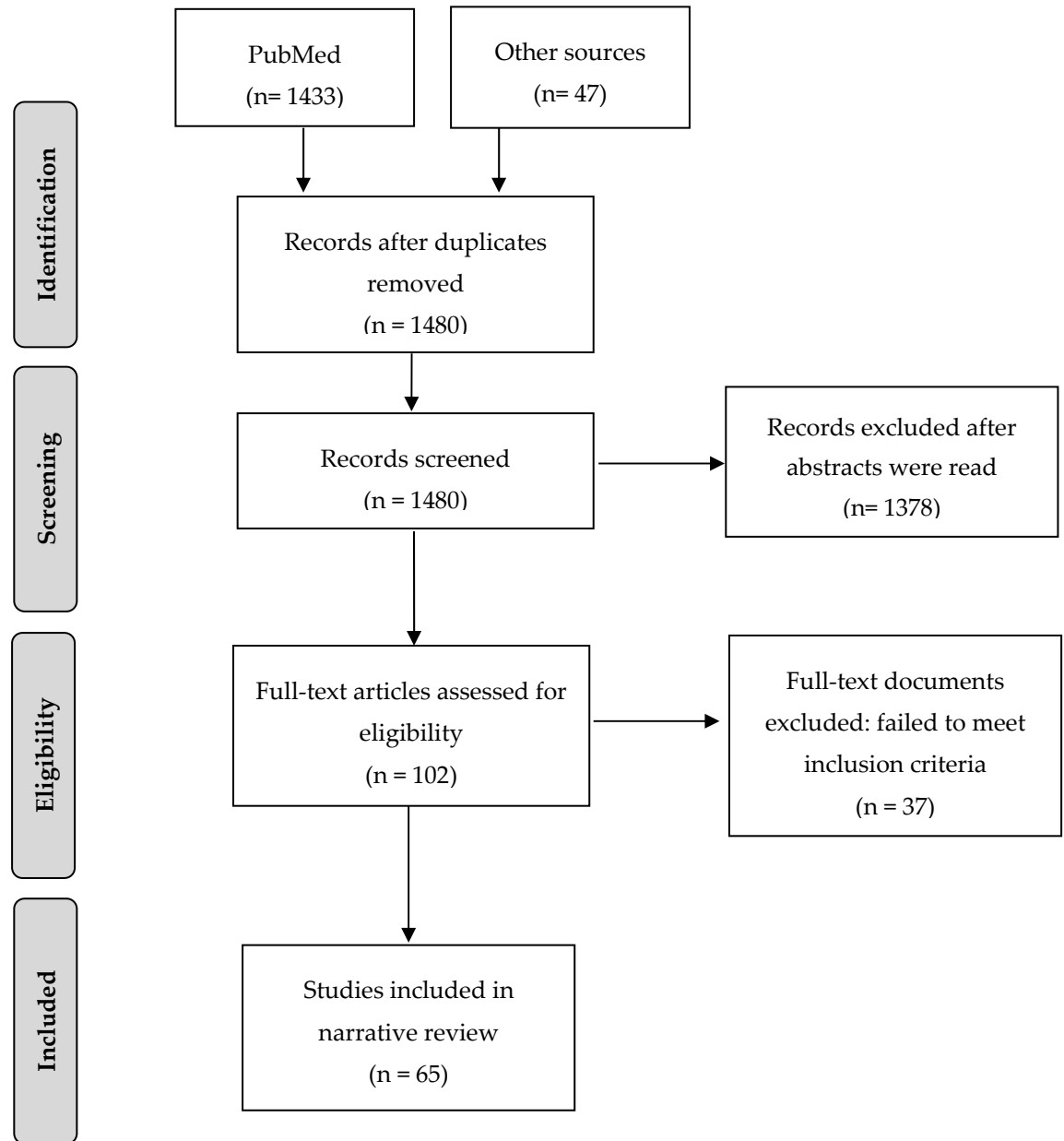

**Figure 1.** Flowchart for studies inclusion.

## 3. Gender Differences in Delusional Disorder

Perhaps because gender differences in DD have been under-investigated, debate has arisen with respect to demographics [9]. While DSM-5 [10] reports no major gender differences in the prevalence or incidence of DD, study results depend on the study design and methodology used. In other words, the estimate of the prevalence or incidence of DD is influenced by the setting from which the sample is recruited (prison, community, inpatient units) and which diagnostic criteria were applied (e.g., prospective, retrospective) [9].

Several authors have reported that DD, in elderly populations, is more often found in women than in men, but this has not invariably been replicated [11,12].

There are several subtypes of DD. The persecutory subtype is the most frequent, but this fact, too, depends on sample characteristics and the setting of the investigation [12–14]. With regard to gender differences in the content of the expressed delusions, several studies have reported erotomanic delusions to be more common in women than in men [14], whereas others have not found any

statistically significant gender differences in the content of delusions. Table 1 shows DSM-5 criteria for DD and subtypes of DD according to the content of delusions.

**Table 1.** DSM-5 criteria and subtypes of DD.

| DSM-5 Criteria for DD | |
|---|---|
| Diagnostic criteria | Existence of one or more delusions that last for at least 1 month. |
| Exclusion criteria | - Criterion A for schizophrenia is never met<br>- Functioning not obviously impaired<br>- Manic or major depressive disorders brief<br>- Delusions cannot be attributed to the physiological effects of a substance or a medical condition |
| **Subtypes of DD according to delusional content** | |
| Persecutory type | Belief that one is being persecuted or conspired against. |
| Jealous type | Conviction that one's lover is unfaithful. |
| Erotomanic type | Belief that someone of relatively high social status is romantically interested. |
| Somatic type | Conviction that one's body is infested, malformed or ill without possibility of cure. |
| Grandiose type | Belief that one is superior to others for several reasons. |
| Mixed type | False beliefs that are a mixture of the above. |
| Unspecified type | Vagueness in the expression of delusions that does not permit classification. |

Clinical variables, not only that of age, but also in the use of substances and in the extent of depressive comorbidity, may impact both the occurrence and the expression of symptoms. Comorbid mood disorders are frequent in patients with DD [13]. Roman-Avezuela and collaborators (2015) reported than women were more likely than men to present with comorbid depressive symptoms [14]. However, this is not consistent with the results of Wustmann and colleagues who found no gender difference in depression in DD [12].

The vast majority of studies report that DD is a stable and valid clinical entity with a remarkable diagnostic stability over time [15], with percentages of stability varying between 64% and 94%. Marneros and colleagues reported a switch to other diagnoses to occur in nearly 21% of cases [16], with no difference between men and women. Gender differences in the association between DD and subsequent dementia have been considered but have not been adequately investigated. Almeida and collaborators (2019) conducted a prospective study of 37,770 men with DD; 8068 (21.4%) developed dementia at follow-up [17]. The study did not, however, include women.

While DD has been classically considered a difficult-to-treat psychotic disorder, antipsychotic medication is nevertheless the treatment of choice [18]. These drugs have shown effectiveness, but more so when combined with non-pharmacological approaches [19,20]. Skelton and collaborators [19], in their systematic review, attempted to evaluate the effectiveness of medication and adjunctive psychotherapy versus adjunctive placebo in patients with DD. They found only one randomized clinical trial addressing this question. It was a study of 17 individuals with DD treated with psychotropics plus either cognitive-behavioral therapy (CBT) or supportive psychotherapy of equal duration [21]. This study found that CBT added to the effectiveness of medication in terms of reducing the strength of conviction and preoccupation with delusional material, increasing insight, and a weakening of the tendency to maintenance and systematization of the delusion. The study did not report gender differences in response to treatment.

With respect to treatment selection, both cognitive ability and psychiatric comorbidity seem to determine the choice of treatment for patients with DD [9,20]. Again, no trials on this issue have considered gender differences.

In the related illness of schizophrenia, women's psychotic symptoms respond at lower antipsychotic doses than men's; women also experience more adverse effects of medication [22]. This finding has been explained in the context of the estrogen protection hypothesis that postulates that high levels of the hormone, estrogen, protects women against psychosis. The hypothesis is based

on the observation that premenopausal women diagnosed with schizophrenia have better clinical outcomes than men, but that this changes after menopause [22]. In DD, findings on gender difference in antipsychotic response are controversial while research studies are few. Some studies report no gender differences in response to antipsychotic medications [15] whereas other findings show that DD in women is more chronic than it is in men [12], suggesting either inferior response, inadequate treatment, or women's failure to adhere to treatment. Comorbid psychiatric diagnoses, such as substance use and depression need to be controlled in order to properly study this question. Adherence to the study protocol has to be carefully monitored because patients with DD have the reputation of not complying with their prescribed regimens [23]. Because many women with DD are postmenopausal, one should probably not expect to find gender differences. At menopause, estrogens levels decline significantly, eliminating the neuroprotection conferred during the reproductive years [22]. Hormonal treatments for women may theoretically influence the incidence of DD in the postmenopausal group. This has, however, not been adequately investigated while treatment adherence was controlled. [3,23].

Menopause is a time when screening for emerging illness (breast cancer, osteoporosis, diabetes, cardiovascular problems) in aging women becomes essential. In schizophrenia, preventive health screening has been found to be lower than in the general population [24]. For DD, we propose several main targets of focused attention in women at the time of menopause (Table 2). Direct treatment of psychotic symptoms is important, followed by the prevention and early intervention in cognitive and other psychopathological comorbidities (mood and anxiety symptoms). Gynecological screening and engagement with other relevant medical specialties must also be a focus in optimal care delivery for women with DD.

**Table 2.** Strategies of care for menopausal women with delusional disorder.

| Health and Social Needs | Strategy |
|---|---|
| **Women's Mental Health** | |
| Psychotic symptoms | Antipsychotics +/- psychological interventions (CBT) |
| Depressive symptoms | Antidepressants +/- psychologic interventions |
| Cognitive symptoms | Early detection; referral to memory clinics; psychological/educational approaches (cognitive remediation); cognitive enhancers |
| **Physical comorbidities** | |
| Hypertension, diabetes mellitus and obesity-cardiovascular disease risk factors, autoimmune disorders | Regular, routine, B.P., BMI checks, glucose, EKG checks, regular GP visits Attention to diet, smoking, exercise, sleep |
| Cancer | Regular screening and GP visits |
| Osteoporosis | Bone density measurements, assessment of fracture risk |
| **Social Needs** | |
| Low income, social adversity | Referral to social work, budgeting, meal preparation Housing, marital support, parenting support, household help Employment support |
| Barriers to help-seeking | Ensuring continuity of care, GP access, follow up of missed appointments; provision of disability income support and childcare |

## 4. How Gender Influences Health Outcomes in Delusional Disorder

The health of men and women is influenced by biological, physiological, and hormonal givens, but also by the life roles individuals assume, their lifestyles, expectations, and responsibilities, all of which directly influence exposure to toxins, to diets, sleep and exercise schedules, to employment conditions, and to daily stressors [2]. These are all impacted by gender [25]. For example, men suffer from heart disease more frequently than women (39% vs. 27%, respectively) [26]; Parkinson's disease is 1.5 times more frequent in men than in women [27]; kidney stones and pancreatitis are also more frequent in men [28]. On the other hand, women suffer more cerebral strokes than men, are more prone to osteoporosis (almost 80% of Americans with osteoporosis are women) [29]; have more urinary tract disease [28], as well as Alzheimer's disease, migraines and multiple sclerosis [30].

Antipsychotics can contribute to health problems. Women taking antipsychotics are at high risk of developing hyperprolactinemia, which results in menstrual irregularities, acne, and hirsutism. In addition, because of hyperprolactinemia, over 50% of men and 30% of women have been shown to experience sexual dysfunction during conventional antipsychotic treatment [31]. Reduced bone mineral density is another common sequela (57% men and 32% women treated with antipsychotics for more than 10 years have hyperprolactinemia) [32,33]. Metabolic syndrome (increased blood pressure, high blood sugar, excess body fat around the waist, and abnormal cholesterol or triglyceride levels), which is seen more frequently in women than men [34], is associated with the development of other pathologies, such as coronary heart disease (which, however, remains more frequent in men), and diabetes mellitus (which occurs more commonly in women).

Despite the dearth of research, we know there is a higher incidence of Alzheimer's disease among women with DD compared to men [23,35]. We know that there is a significantly higher prevalence in men than in women of alcohol use disorder and abuse of other substances [23] although, thus far, this has been reported mainly in the delusional infestation form of DD [36]. Comorbidity of substance use can lead to accidents, trauma, aggression, suicide, and acts of violence. Screening for sex-specific corollaries of illness will result in better care (Table 3).

**Table 3.** Gender influence on health outcomes.

| Men | Women |
|:---:|:---:|
| **Gender influence on health outcomes in general population** | |
| Heart disease<br>Parkinson's disease<br>Kidney stones<br>Pancreatitis | Osteoporosis<br>Urinary tract problems<br>Alzheimer's disease<br>Migraine<br>Multiple sclerosis |
| **Gender influence on health outcomes in schizophrenia** | |
| Sexual dysfunction<br>Coronary heart disease | Menstrual irregularities<br>Reduced bone mineral density<br>Metabolic syndrome<br>Diabetes mellitus<br>Autoimmune disorders |
| **Gender influence on health outcomes in delusional disorder** | |
| Cardiovascular disease<br>Alcohol use | Alzheimer's disease<br>Osteoporosis |

## 5. Health Outcomes in Women with Delusional Disorder

In the related illness of schizophrenia, reduced estrogen levels at the time of menopause means the loss of the neuroprotection conferred by estrogens [37]. The postmenopausal stage of schizophrenia requires special therapeutic attention as these women show a worsening of psychotic symptoms, require higher antipsychotic doses [9,38] and present with greater severity of depressive symptoms at this time [39]. DD starts at this age [9], when psychotic symptoms, antipsychotic adverse effects, and comorbid conditions all increase. Table 4 summarizes some of the metabolic and other physical disturbances that should be taken into account when focusing on health outcomes in women with DD.

**Table 4.** Recent physical health studies in women with DD.

| Metabolic Disturbances and Cardiovascular Disease | |
| --- | --- |
| **Observation** | **Potential explanation** |
| Increased adult adiposity, insulin sensitivity and blood lipid levels | Metabolic syndrome may be attributable to reduce estrogen levels at menopause and to the use of antipsychotics |
| Increased cardiovascular risk | Occurs at advanced age. Loss of estrogens at menopause, lifestyles (smoking and high BMI) and antipsychotics use also contribute |
| **Neurologic disorders** | |
| **Observation** | **Potential explanation** |
| Increased risk of cognitive disorders (or dementia) | Patients with DD show impaired verbal memory and other cognitive symptoms attributable to an increase in cerebrovascular events (because of age and the use of antipsychotics) |
| Movement disorders (extrapyramidal symptoms and tardive dyskinesia) | Loss of estrogens at menopause increases its risk of tardive dyskinesia. Antipsychotic dose may be too high. The incidence of extrapyramidal disorders: increases with age |
| **Autoimmunity** | |
| Autoimmune diseases | Women are more susceptible to autoimmune disorders than men. At menopause, the risk is increased |
| **Risk of cancer** | |
| Gynecological cancers | Women with DD may show low compliance with gynecological appointments and are less likely than peers to receive cancer screening |
| Other cancers | Little is known. Lifestyle factors may contribute to an increased mortality. |

*5.1. Metabolic Disturbances and Cardiovascular Disease in Women with Delusional Disorder*

Nevertheless, antipsychotics are the treatment of choice in DD as they are in schizophrenia [20], so that antipsychotic-related adverse events can be expected to increase at menopause.

Although there are several contributory factors to adiposity, postmenopausal women show increased abdominal fat, increased plasma levels of triglycerides and total cholesterol compared to premenopausal women [40,41], all risk factors for cardiovascular disease. The use of antipsychotics at this period of life increases the risk of both metabolic syndrome and cardiovascular disease [22,42]. An important adverse and potentially very serious effect of antipsychotic medication is the prolongation of the QTc interval on the ECG [43]. Women are more vulnerable to this effect than men [44].

Although cardiovascular risk increases with advanced age in both sexes, the remodeling of the heart and blood vessels that takes place in middle age differs between sexes [45]. In contrast to women, men are more likely to present with heart failure [42,46,47]. Older men are more likely to present with heart failure with reduced ejection fraction because the number of ventricular myocytes suffer a reduction with age in men, but not in women [47]. Cardiac ageing in women is more characterized by an increase in diastolic dysfunction.

Recent evidence suggests that 10% of women experience premature or early-onset menopause, at around the age of 45, which heightens their risk for cardiovascular disease and mortality [48]. Smoking and high body mass index, common in individuals with psychotic illness, further increases the risk [49]. It would be interesting to investigate whether women with DD experience menopause at a relatively early age.

*5.2. Neurologic Disorders in Delusional Disorder*

Postmenopausal loss of estrogens leads to an increase in the prevalence of cognitive dysfunction, cerebrovascular events and extrapyramidal effects [22]. Neurologic manifestations of aging also include an increase in the prevalence of Parkinson's and Alzheimer diseases. Female sex and post-menopausal status are risk factors for late-onset Alzheimer's disease (AD) [50] while women with early menopause are at special risk [51]. Recent studies have shown that patients with DD present impaired verbal memory and other cognitive symptoms which suggests the need for cognitive interventions [20,52].

Patients with very-late onset DD have an approximately eight-fold increased risk of dementia [53]. These are results from a nationwide register of patients in Denmark, both in- and out- patients who were compared to the general population and to patients with osteoarthritis [53]. Very late first-contact delusional disorder patients showed an increase rate of dementia compared to very late first-contact osteoarthritis patients.

Parkinson's disease (PD), by contrast to Alzheimer's, is more common in men [54]. Both men and women with DD suffer from extrapyramidal symptoms and tardive dyskinesia, attributable to the use of antipsychotics [55,56]. A recent systematic review of studies of tardive dyskinesia shows that prevention is more effective than treatment [56]. The authors recommend limiting the prescription of antipsychotics by using the minimum effective dose and minimizing the duration of therapy. If discontinuation of antipsychotics is not possible, switching from a first-generation to a second-generation antipsychotic lowers D2 affinity and reduces the risk of tardive dyskinesia symptoms. With respect to treatment, deutetrabenazine and valbenazine have the strongest evidence of efficacy.

Recent work has provided evidence of brain abnormalities in the frontal and cingulate cortex and insula of patients with DD [57]. A review on this topic reported that structural and functional brain imaging in patients with DD reveals that some patients present signs of lacunar infarcts in white matter, frontal and temporoparietal lobes or lacunar infarcts in basal ganglia [58]. This may be important because treatment non-response has been related to the presence of such brain lesions. No gender differences have been reported.

*5.3. Autoimmune Diseases in Delusional Disorder*

Women are, in general, significantly more susceptible to the development of autoimmune diseases than men [59] and many of these diseases, especially systematic lupus erythematosus, tend to be associated with schizophrenia [60]. Patients with schizophrenia have a 50% increase in the lifetime prevalence of one or more autoimmune disorder [60] and this may also be true for patients with DD [61,62]. In fact, earlier work has postulated an elevated incidence of human leukocyte antigen (HLA) class I alleles in patients with schizophrenia and DD, suggesting a shared biological vulnerability.

*5.4. Risk of Cancer in Delusional Disorder*

The prevalence and risks for cancer has not been adequately investigated in women with DD. A case report and review of the literature on DD and oncology [63] highlighted the fact that DD is an under-researched condition with little information currently available about the risk of comorbid malignancies. There are many barriers to regular cancer screening (colon, cervix, breast) in individuals with psychotic illness [64]. Promotion of screening is vital in this population because, while incidence rates of cancer are the same as in the general population, mortality rates are much higher [65,66]. This is partly due to late diagnosis [67] and draws attention to the need for closer collaboration between psychiatry and other medical disciplines [68].

## 6. Social Needs in Women with DD

According to DSM-5, DD is considered to be a psychotic disorder without significant functional impairment [10]. Despite this, social circumstances and treatment consequences need to be carefully addressed in this population [10]. Shared psychotic disorder is an example of dysfunction on the social dimension. Christensen and Ramos reported the case of a 26-year old man who lived with his 63-year-old mother and 84-year-old father [69]. The family lived in a state of such poverty and disadvantage that it ultimately led to the development of persecutory and grandiose delusions. Therapeutic options were limited by the gravity of the social needs.

It is uncommon for family members to share the delusions of DD patients but they are usually very distressed by their inability to deal with the unshakeable false beliefs of such patients. González-Rodríguez and Seeman [9] have suggested that mental health staff model psychosocial

interventions that work so that family members can master them and reduce tensions at home. More rigorous investigation into effective psychosocial interventions need to be conducted in the context of DD.

## 7. Discussion

Research supports the observation that gender has a profound impact on the expression of mental disorders. Although research in DD is sparse, women appear to suffer from a somewhat different health burden than men-more depression and cognitive disorders, different antipsychotic adverse effects, more autoimmune disease, and sex-specific psychosocial needs.

Patient engagement with services and patient-led improvement of services have been internationally recognized as a quality indicators of healthcare systems, but few studies on this topic exist that bear a direct relation to DD [70]. We were impressed by a comprehensive meta-ethnography that synthetized information on how the nurse-patient relationship in particular can enhance patients' engagement and patients' health in community care settings [71]. The strength and continuity of that relationship can mediate patients' well-being physically, emotionally, and socially [71]. On that basis, we propose, as a service goal, the forging of a strong patient-nurse bond, which, we hypothesize, would increase compliance rates with medication and promote referral to needed medical/surgical specialties. Patients' adherence to recommended medications are quality indicators of healthcare [72] and, while no single intervention strategy is sufficient, the nurse-patient relationship has been found essential to this task [72,73]. The nurse-patient relationship is key to encouraging and maintaining the patient's role in her (or his) own recovery [73,74]. It has the potential of ensuring attendance at primary care visits and compliance with essential health screening. A strong bond of trust between patient and nurse helps in the reduction of tobacco use and substance abuse. It also helps to prevent sexual risk behaviors. Nurses are also active agents in forging links with social work to obtain adequate housing and disability incomes for impoverished patients [72,73]. Recent work has systematically reviewed measures available to improve important aspects of the nurse-patient relationship [74]. Although a substantial amount of existing tools was reported, their quality still needs to be assessed [74]. While therapeutic relationships need to be forged with all members of a treatment team for medical care to be successful, the nurse-patient relationship seems to us to be key for a variety of reasons that have previously been addressed in the literature [75,76]. Patients with DD are slow to trust, so that this relationship may take time and endurance, but will pay numerous dividends.

## 8. Conclusions

DD is a difficult-to-treat clinical condition with sex-specific health needs. This narrative review has addressed important questions focused on the treatment of women with DD, with special attention to sex-specific health needs. DD is an illness that most often first occurs in middle to late adult life, approximately the time of menopause in women. Menopause and age, together, increase the severity of many health problems in women. In fact, age at menarche, reproductive years and menopausal status have been correlated with adult adiposity, insulin sensitivity and blood lipid levels. The use of antipsychotics at this period of life also increases the risk of metabolic syndrome, which is not only attributable to menopause itself, but also to antipsychotic use. This is perhaps why cardiovascular risk increases at menopause. Postmenopausal loss of estrogens leads to an increase in the prevalence of cognitive dysfunction. Patients with DD present impaired verbal memory and other cognitive symptoms, a fact that may increase should cerebrovascular events occur during the postmenopausal period. Studies investigating family history of autoimmune disorders have shown an association with psychotic disorders such as DD. Women with schizophrenia are less likely to receive mammograms or pelvic examinations compared to healthy women. This may also be the case for women with DD although it has not been specifically investigated. When focusing on the social needs in women with DD, social circumstances and treatment consequences appear to be linked and should be carefully addressed. A gender-specific approach to individuals with DD may be able to reduce the impact of

comorbid illness in this condition. We propose that successful interventions at the patient, provider and system levels can be mediated by a strong, ongoing nurse-patient relationship.

## 9. Future Perspectives

The observation that gender has an impact on health outcomes in patients with DD has been widely replicated in recent years. Future studies should be focused on the investigation of gender differences in social, economic and biological determinants of physical health and DD, as many ethiopathological features appear to reinforce each other. Men are more vulnerable to chronic diseases such as coronary heart disease, cancer and cerebrovascular disease compared to women. However, at the time of menopause when estrogen protection is lost women may also suffer from these conditions. Men and women show different responses to drugs, partly due to physiology and partly to lifestyle. Research is needed in the field of gender differences in DD. Questions that need to be addressed are why women are less likely than men to receive screening for cancer. At the time of menopause, many women's level of stress is considerably increased by the assumption of a double caretaking role for young adult children and elderly parents. This may lead to a neglect in self-care. Future research into relationships that effectively alleviate stress (such as nurse-patient relationships) is an important target that will improve the planning of care services for women with DD.

**Author Contributions:** A.G.-R., M.V.S. and A.Á. were involved in the electronic search and selection of papers and wrote the first draft of the manuscript. A.G., N.S. and G.F.F. supported data and collaborated in the first draft of the manuscript. M.V.S. and D.J.P. critically revised the manuscript. J.L. revised the paper and supervised all the review. All authors have read and agreed to the published version of the manuscript.

**Funding:** This research did not receive any specific grant from funding agencies in the public, commercial, or not-for-profit sectors.

**Acknowledgments:** J.L. received an Intensification of the Research Activity Grant (SLT006/17/00012) from the Health Department of the Generalitat de Catalunya.

**Conflicts of Interest:** J.L. has received honoraria for lectures or advisory board membership from Janssen, Otsuka, Lundbeck and Angelini. A.G.R., A.A. and A.G.D. have received honoraria or paid for travels from Janssen, Otsuka, Lundbeck and Angelini.

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
