# Peer review of "Care for Women with Delusional Disorder: Towards a Specialized Approach"

_women, doi:10.3390/women1010004_

Round 1
Reviewer 1 Report
Dear Authors,
Congratulations on the work presented. Below, I will give you a few pointers to take into account, with the sole purpose of trying to improve this original that, based on my opinion, would be publishable with only a few minor changes.
- The article presents the results of a data analysis on specialised literature but does not detail the review protocol used, the eligibility criteria of the articles analysed, or the methods used to extract the data. Although it is not a systematic review, the methodological procedure followed should be clarified a bit more. The work would benefit from an explanatory table or flowchart in this regard.
- A more robust review of the most recent findings is needed, taking into account current lines of research.
- After a well-founded study that collects interesting ideas, the conclusions are somewhat lacking. They could be redrafted to highlight the main findings of the study in a more decisive way. The same can be said of the possible proposals for future lines of research. This section could be strengthened.
I hope these suggestions will help you to improve the study and that you will finally see it published in this journal.
Author Response
REVIEWER 1
- The article presents the results of a data analysis on specialised literature but does not detail the review protocol used, the eligibility criteria of the articles analysed, or the methods used to extract the data. Although it is not a systematic review, the methodological procedure followed should be clarified a bit more. The work would benefit from an explanatory table or flowchart in this regard.
We entirely agree with Reviewer 1 that the review protocol should have been better described. We have now described, in detail, the methodological procedure and have built a flowchart for better clarification.
- A more robust review of the most recent findings is needed, taking into account current lines of research.
We agree with reviewer one that most of the recent findings should have been better described. We have now included a new Table 4 that summarizes the results.
The paragraphs we have added are marked in yellow in section 5 and in all subsections.
- After a well-founded study that collects interesting ideas, the conclusions are somewhat lacking. They could be redrafted to highlight the main findings of the study in a more decisive way. The same can be said of the possible proposals for future lines of research. This section could be strengthened.
We entirely agree with Reviewer 1 that the conclusions section was overly short, and the conclusions too sparse. We have expanded this section and added a paragraph on future perspectives.
Reviewer 2 Report
Overall summary
The authors conducted a narrative review to analyse specific health needs of women with delusional disorder (DD). Likewise, the authors evaluated how to integrate the management of these patients into service programs.
General comments
The topic is interesting and in general, the manuscript is clear, and it is quite well written. Likewise, although several self-citations are included (#7-9, 22, 23, 31, 38, 39, 58, 60, 64), I think they should be continued as they show expertise in the study area of the authors.
However, as it is noted in detail below, there are a few concerns (particularly related to the section of “methods”) that should be addressed prior to this paper is considered for publication.
- Line 36. The authors must review the reference #3, because the higher prevalence of eating disorders in girls is not included in it.
- Lines 52-55. The authors must include a reference for the sentence: “Because DD is a… women of different ages”.
- Lines 68-76. As the authors know, the section of Materials & Methods must provide sufficient detail to allow the work to be reproduced. According to this, more details about the methodology carried out in the present narrative review are necessary. At least, the authors must include keywords used in the searches as well as their combination using Boolean operators, and filters used (eg. types of study designs). Likewise, a flow diagram for the strategy of searching for the studies would be very useful to understand how “several hundred articles” were discarded and only 65 of them were identified as relevant to the present review.
- Line 68. Why was the section of methods (subsection 1.2.) included within the section of introduction?
- In relation to the section of “gender differences in DD”: Could the hormonal treatments for postmenopausal women influence the incidence of DD in this group?
- Lines 81-82. How the study design and the methodology used influence the data about differences in the prevalence or incidence of DD should be explained.
- Section of references must be reviewed. There are a lot of mistakes (eg. ref. #18 “Ddisorder”, ref. #45 “Mmn”, ref. #48 “Ttme”, ref. # “Dpecific”, ref. #46 “geart”…).
Other comments
- In the main text, the abbreviations only must be clarified once (eg. DD was clarified 4 times: in lines 52, 71, 79 and 280). Likewise, from the moment an abbreviation is clarified, it must be used in the rest of the manuscript (eg. change “delusional disorder” by “DD” in lines 62 and 284).
- In the table 1: i) “Osteoporosis” and “BMI checks” must not be included in the same cell; ii) please delete the space which is after “childcare” and before “,”.
Author Response
REVIEWER 2
1.Line 36. The authors must review the reference #3, because the higher prevalence of eating disorders in girls is not included in it.
We have removed this reference and now offer a concise prevalence of gender differences in eating disorders and a review on sex and gender differences in mental disorders (Lancet, 2016).
2.Lines 52-55. The authors must include a reference for the sentence: “Because DD is a… women of different ages”.
We have included new references (3 and 8). Both are papers reviewing gender differences in psychosis, which are outstanding contributions to this field.
3.Lines 68-76. As the authors know, the section of Materials & Methods must provide sufficient detail to allow the work to be reproduced. According to this, more details about the methodology carried out in the present narrative review are necessary. (..). Likewise, a flow diagram for the strategy of searching for the studies would be very useful to understand how “several hundred articles” were discarded and only 65 of them were identified as relevant to the present review.
We agree with Reviewer 2 that a more detailed description of methodology should have been provided. We now describe the search strategy and have built a flowchart.
4.Line 68. Why was the section of methods (subsection 1.2.) included within the section of introduction?
We entirely agree with Reviewer 2 that the methods section should be separated from the introduction. Given this change, we have renumbered all sections.
5.In relation to the section of “gender differences in DD”: Could the hormonal treatments for postmenopausal women influence the incidence of DD in this group?
Hormonal treatments for postmenopausal women may theoretically influence the incidence of DD in this group, however, adherence to somatic treatments has not been adequately investigated. This question will be better addressed in future studies designed to specifically investigate the influence of various treatments on the incidence of DD. This is a very good question. We have addressed this in the “gender differences” section.
6.Lines 81-82. How the study design and the methodology used influence the data about differences in the prevalence or incidence of DD should be explained.
We have included a paragraph to better clarify this point. The estimation of the prevalence or incidence of DD may be influenced by the setting where the sample was recruited (prison, community, inpatient units) and which criteria were applied to diagnose patients with DD (prospective, retrospective, etc) [9].
7.Section of references must be reviewed. There are a lot of mistakes (eg. ref. #18 “disorder”, ref. #45 “Mmn”, ref. #48 “Time”, ref. # “Specific”, ref. #46 “geart”…).
We have corrected all these mistakes. Changes have been marked in yellow in the reference section.
Other comments
1.In the main text, the abbreviations only must be clarified once (eg. DD was clarified 4 times: in lines 52, 71, 79 and 280). Likewise, from the moment an abbreviation is clarified, it must be used in the rest of the manuscript (eg. change “delusional disorder” by “DD” in lines 62 and 284).
We have corrected all mistakes regarding abbreviations.
2.In the table 1: i) “Osteoporosis” and “BMI checks” must not be included in the same cell; ii) please delete the space which is after “childcare” and before “,”.
We have corrected all these issues. We have noted strategies for the assessment of osteoporosis.
Reviewer 3 Report
In the present review, the Authors aimed to cover a gap in the clinical/academic literature by focusing on the specific health needs of women with Delusional Disorder (DD). At the end, They propose a program of service that focuses on the comprehensive care of women with DD.
Overall, I found the present Review timely, well conducted, very interesting and scientifically sound. I have only some minor comments aimed to improve the high quality of the paper and these are outlined below:
1) I believe that a table with DSM-V criteria and types of DD should be inserted as it may be useful to the reader.
2) Moreover, a table with the more relevant papers included in the Review would be useful.
Author Response
REVIEWER 3
1.I believe that a table with DSM-V criteria and types of DD should be inserted as it may be useful to the reader.
We have built a new Table 1 that presents DSM-5 criteria for DD as well as a definition of the subtypes of DD.
2.Moreover, a table with the more relevant papers included in the Review would be useful.
We added a new Table 4 with a summary of relevant findings in the field of DD and health outcomes.